# What We Miss Matters: Learning from the Overlooked in Point Cloud Transformers

**Yi Wang**[1,*] **Jiaze Wang**[2,3,*] **Ziyu Guo**[2]**, Renrui Zhang**[2]**, Donghao Zhou**[2]**,**

**Guangyong Chen**[4]**, Anfeng Liu**[1,†] **Pheng-Ann Heng**[2]

[1] Central South University [2] The Chinese University of Hong Kong
[3] FitX Technology (Hong Kong) Limited [4] Zhejiang Lab

## Abstract

Point Cloud Transformers have become a cornerstone in 3D representation for their ability to model long-range dependencies via self-attention. However, these models tend to overemphasize salient regions while neglecting other informative regions, which limits feature diversity and compromises robustness. To address this challenge, we introduce **BlindFormer**, a novel contrastive attention learning framework that redefines saliency by explicitly incorporating features typically neglected by the model. The proposed Attentional Blindspot Mining (ABM) suppresses highly attended regions during training, thereby guiding the model to explore its own blind spots. This redirection of attention expands the model's perceptual field and uncovers richer geometric cues. To consolidate these overlooked features, BlindFormer employs Blindspot-Aware Joint Optimization (BJO), a joint learning objective that integrates blindspot feature alignment with the original pretext task. BJO enhances feature discrimination while preserving performance on the primary task, leading to more robust and generalizable representations. We validate BlindFormer on several challenging benchmarks and demonstrate consistent performance gains across multiple Transformer backbones. Notably, it improves Point-MAE by +13.4% and PointGPT-S by +6.3% on OBJ-BG under Gaussian noise. These results highlight the importance of mitigating attentional biases in 3D representation learning, revealing BlindFormer's superior ability to handle perturbations and improve feature discrimination. Project page: https://winfred2027.github.io/projects/BlindFormer/intro.html.

## 1 Introduction

Point clouds serve as a fundamental 3D representation with broad applications in robotics [6, 47], autonomous driving [7], augmented reality [3], and virtual reality [13]. While their versatility highlights the need for robust space understanding, the irregular and sparse nature of point clouds complicates efficient processing.

The recent success of Transformer architectures [34, 5, 65] in point cloud analysis has demonstrated remarkable capabilities in capturing long-range dependencies through self-attention mechanisms. These models follow a paradigm where the attention weights automatically emphasize salient regions for understanding the point cloud while downplaying less significant areas. Originally designed for natural language, attention mechanism has been successfully adapted for 2D vision. However, unlike natural language [9] or images [17], which often contain redundant information such as

---

*These authors contributed equally to this work.
†Corresponding author: afengliu@csu.edu.cn

39th Conference on Neural Information Processing Systems (NeurIPS 2025).

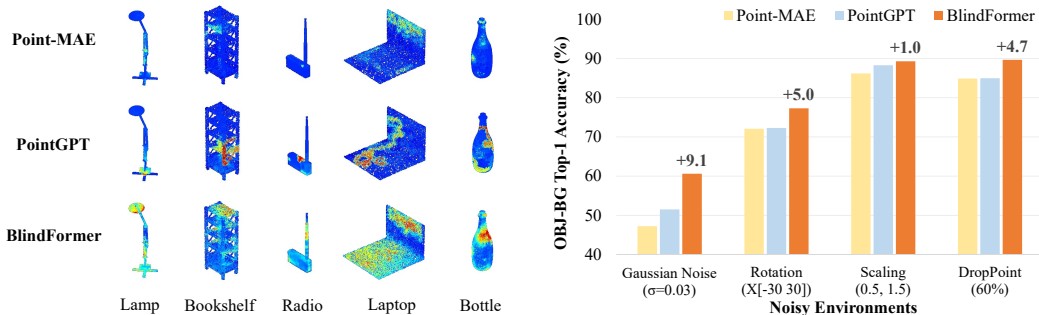

Figure 1: **Illustration of BlindFormer's Advantages.** Point-MAE is employed as the backbone of our proposed BlindFormer. **Left:** BlindFormer emphasizes extracting information from a greater number of patches. **Right:** BlindFormer demonstrates greater robustness than previous methods.

contextual structures and backgrounds, point cloud data are inherently sparse, meaning that each point or region is critical to the overall representation. This scarcity of redundant information implies that Transformer-based models, when neglecting less prominent point patches, may inadvertently overlook essential latent information. This observation leads to a pivotal question: *Can we design a framework that actively mitigates attentional blind spots to improve both robustness and feature discrimination in 3D representation learning?*

To answer this question, we re-examine the attention weights in Transformer-based point cloud models. As illustrated in Figure 1, we find that models like Point-MAE [65] and PointGPT [5] primarily rely on a limited set of high-attention patches for analysis. This reliance presents two significant issues: **(1)** Increased sensitivity to perturbations. Over-focusing on high-attention patches makes the models more susceptible to noise and incomplete data, as disturbances in these areas disproportionately affect performance. **(2)** Inadequate feature discrimination. Ignoring potential information in low-attention regions limits the model's ability to distinguish objects with similar local structures, reducing their ability to generalize across distribution shifts.

To address the limitations of attentional bias, we introduce **BlindFormer**, a contrastive attention learning framework that can be seamlessly integrated into existing point cloud Transformers. Our approach comprises two key components: First, we introduce an Attentional Blindspot Mining (ABM) module, which reshapes the attention distribution by explicitly suppressing salient regions during training. By constructing a masking distribution derived from self-attention scores, ABM selectively occludes patches that contribute most to the global representation. This encourages the model to redirect its focus toward previously overlooked blind spots, thereby learning to capture spatial structures from more subtle cues and expanding its perceptual field. To reinforce these under-attended features, BlindFormer further incorporates Blindspot-Aware Joint Optimization (BJO), a united learning objective that integrates blindspot feature alignment with the original pre-training task. This joint supervision not only retains the model's task-specific capabilities but also enhances its ability to capture invariant and discriminative features across diverse point clouds.

Extensive experiments demonstrate that our BlindFormer significantly enhances the robustness of Transformer-based models across various noisy environments, including Gaussian noise, rotation, scaling, and point dropout. Specifically, under Gaussian noise perturbation, BlindFormer improves the classification accuracy of Point-MAE by 17.2% in the OBJ-ONLY setting. Similarly, under rotation perturbation, it boosts the segmentation performance of PointGPT-S by 3.3% on ShapeNetPart. Furthermore, BlindFormer achieves state-of-the-art performance across various downstream tasks for 3D understanding. Even when competing models are given additional training time, BlindFormer maintains a consistent performance advantage. These results highlight BlindFormer's potential to effectively address the limitations of existing Transformer-based models by comprehensively activating regions and improving feature discrimination capabilities.

Our main contributions can be summarized as follows:

**(I)** We propose BlindFormer, a novel contrastive attention learning framework for point cloud understanding that explicitly addresses attentional blindspots which enhances the model's ability to capture global structures and significantly improves its robustness and generalization capabilities.

**(II)** We introduce an attentional blindspot mining strategy that dynamically suppresses dominant regions and encourages the model to focus on previously neglected parts, promoting the learning of spatial features from a broader set of patches rather than over-relying on a salient subset.

**(III)** Extensive experimental results demonstrate that BlindFormer can be seamlessly integrated into mainstream transformer architectures and achieve significant improvements across a variety of 3D understanding tasks.

## 2   Related Works

**Self-Supervised Learning for NLP and Image.** Self-supervised learning (SSL) has emerged as a powerful paradigm in natural language processing (NLP) [11, 75] and computer vision [41, 14, 64, 32, 40, 1, 25], enabling models to learn rich representations from unlabeled data. In NLP, BERT [9] exemplifies this by randomly masking input tokens and training the model to predict them, fostering deep contextual understanding. ELMo [45] utilizes bidirectional LSTMs to generate contextualized word embeddings, while GPT [41] adopts an autoregressive approach with a unidirectional Transformer to predict the next word, fine-tuning all parameters for specific tasks. Recently, generative SSL methods have begun to outperform contrastive approaches in computer vision. Masked Autoencoders [17] randomly mask image patches and train the model to reconstruct the missing pixels, leading to effective visual representations. BEiT [4] extends this by tokenizing image patches and predicting masked tokens, integrating NLP techniques into vision tasks. Additionally, Image GPT [28] treats images as sequences of pixels and trains a Transformer to autoregressively predict pixels without explicit spatial structure, demonstrating strong representation learning. This shift towards generative self-supervised learning methods not only demonstrates their ability to capture comprehensive data representations and improve performance in NLP and computer vision but also highlights their significant potential in advancing point cloud processing and analysis.

**Self-Supervised Learning for Point Cloud.** Various methods have been investigated for self-supervised learning on point clouds [54, 27, 57, 69, 70, 16, 52, 66, 12, 58, 56]. Many works focused on generative modeling with generative adversarial networks and autoencoders, aiming to reconstruct inputs using different architectural designs [31, 65, 46, 22, 2, 55, 21, 51, 43]. PointMAE [34] proposes an effective scheme of masked autoencoders for point cloud self-supervised learning. Point-M2AE [67] further employs a hierarchical transformer architecture and implements a specific masking strategy. PointGPT [5] proposes a point cloud auto-regressive generation task to pre-train transformer models. Moreover, contrastive methods also have been extensively explored [39, 61, 62, 33, 72, 60, 19]. DepthContrast [72] generates augmented depth maps and conducts instance discrimination on the extracted global features. MVIF [20] employs cross-modal and cross-view invariance constraints to enable self-supervised learning of modal- and view-invariant features. OcCo [49] aims to reconstruct the original point cloud from an occluded version observed in camera views. Some studies focus on integrating cross-modal information, utilizing knowledge from language or image models to enhance 3D learning [37, 10, 50, 44, 38]. PointCLIP [68] facilitates the alignment between point clouds encoded by CLIP and corresponding text descriptions, enhancing cross-modal understanding. PointCLIP V2 [74] uses a shape projection module to guide CLIP in generating more realistic depth maps and prompts a GPT model to create 3D-specific text for textual encoder input.

## 3   Methods

The overall framework of BlindFormer is illustrated in Figure 2. First, the Attentional Blindspot Mining module generates a masked point cloud by dynamically suppressing regions with high attention scores. Both masked point cloud and the original input point cloud are then fed into the shared backbone model to obtain the global features of each input. By explicitly aligning the features from these two branches, BlindFormer guides the model to better capture and understand information from the overlooked regions, enhancing feature discrimination and generalization. During the pre-training stage, the model is optimized using a combination of blindspot feature alignment and the original pre-training loss—such as the reconstruction loss from Point-MAE [34] or the generation loss from PointGPT [5]. After pre-training, we employ the backbone model without the masking strategy, leveraging the learned latent representations for downstream tasks.

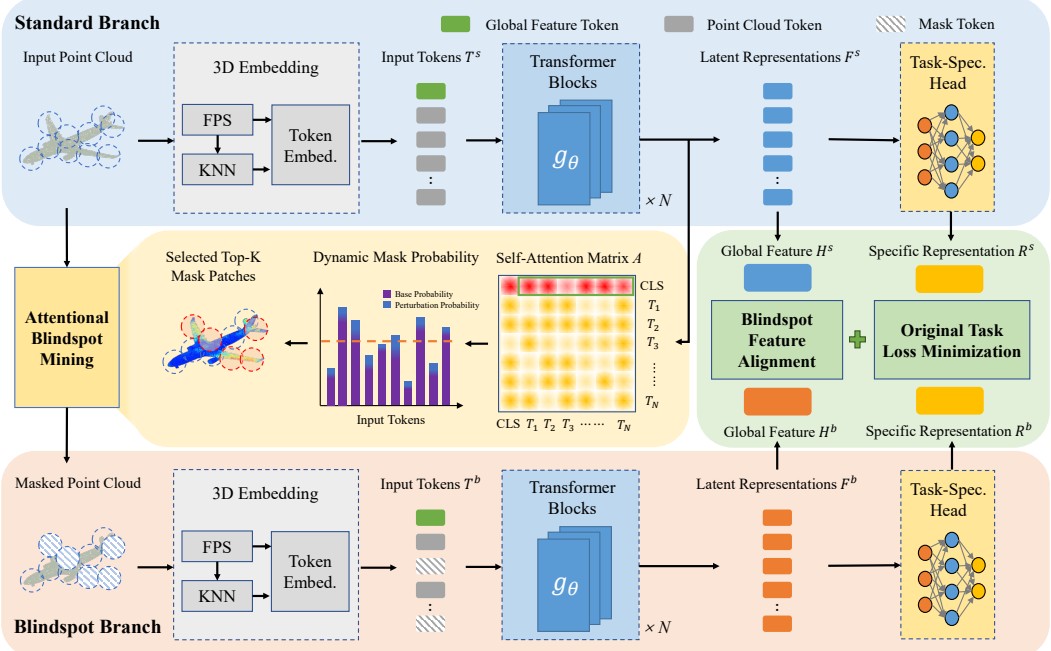

Figure 2: **Overview of the BlindFormer Framework.** BlindFormer consists of two branches that share the same weights: a standard mode branch and a blindspot mode branch. An attentional blindspot mining module generates a masked point cloud by selecting less activated patches from the output of the standard mode branch. Both branches process their respective inputs through the shared Transformer blocks to obtain latent representations. Finally, a blindspot-aware joint optimization is used to align the representations of these two branches.

## 3.1 Attentional Blindspot Mining

To fully leverage the self-attention mechanism and mitigate the model's reliance on a small subset of key patches, we propose an Attentional Blindspot Mining (ABM) strategy, which guides the model to focus on blindspot regions and enforces a more comprehensive understanding of the whole structure in challenging scenarios by dynamically masking salient areas.

**Point patch attention.** Given a point cloud $X \in \mathbb{R}^{p \times 3}$, we utilize Farthest Point Sampling (FPS) and K-Nearest Neighbors (KNN) algorithms to identify $n$ center points $C$ and their corresponding $k$ nearest neighbors, forming $n$ point patches $P$. We employ the self-attention mechanism in the transformer architecture to compute the attention weights of point patches relative to the global feature. A new set of input tokens $T \in \mathbb{R}^{(n+1) \times d}$, consisting of point tokens $T^p \in \mathbb{R}^{n \times d}$ and a learnable global feature token $T^f \in \mathbb{R}^{1 \times d}$, is utilized to compute the queries $Q \in \mathbb{R}^{(n+1) \times d}$, keys $K \in \mathbb{R}^{(n+1) \times d}$, and values $V \in \mathbb{R}^{(n+1) \times d}$. The attention matrix $A$ is subsequently derived from the dot product of queries and keys. Since the first element of the input tokens $T_1$ corresponds to the global feature token, the first row of the attention matrix can be interpreted as the contribution of each token to the global feature. Considering the output tokens depend on both the attention matrix and the values, we incorporate the norm of $V_j$ when determining the significance score of token $j$. Consequently, the attention matrix and significance score for point patch $j$ are computed as follows:

$$A = Softmax(QK^T / \sqrt{d}), \tag{1}$$

$$S_j = \frac{A_{1,j} \times \|V_j\|}{\sum_{i=2} A_{1,i} \times \|V_i\|}, \tag{2}$$

where $i, j \in 2, ..., n+1$. For a multi-head attention layer, we compute the significance scores for each head separately and aggregate them by taking the sum over all heads.

**Dynamic blindspot generation.** A straightforward idea is to mask the top $k$ patches with the highest significance scores, as they are key to the model's understanding of the point cloud. However, a fixed masking probability merely shifts the model's attention without engaging a broader set of

patches. As the model becomes reliant on new areas of focus, it similarly falls into the trap of limited comprehension of the point cloud. Our primary objective is to ensure that high-attention regions have a higher likelihood of being suppressed. Therefore, we propose a dynamic blindspot generation strategy. Specifically, we construct an updatable base masking probability using the latest self-attention significance scores, prioritizing the masking of patches that currently contribute significantly to the global features. Additionally, Gumbel noise[30] is introduced as a perturbation probability, derived from a uniform distribution $U[0, 1]$, to enhance the variability of the masking probability. Based on this concept, the final dynamic masking probability $p_{dy}$ is expressed as:

$$p_{dy} = \log\left(Softmax(S/\tau_{pro})\right) - \log\left(-\log\varepsilon\right), \tag{3}$$

where $\tau_{pro}$ is a temperature hyperparameter which controls the sharpness of the base masking probability. A lower temperature (less than 1) results in a sharper distribution, meaning that regions with the highest attention are more likely to be masked. Based on the dynamic masking probability, we apply simple Top-K strategy to select the $k$ point patches $P^{mask} \in \mathbb{R}^{k \times 3}$ to be masked:

$$P^{mask} = \text{Top-K}(p_{dy}, k), \tag{4}$$

$$P^{blind} = P - P^{mask}. \tag{5}$$

The blindspot region $P^{blind}$ contains the previously under-attended patches, which are preserved as input for the model. In this manner, regions that previously attract high attention are more likely to be suppressed, promoting a deeper understanding of the whole spatial information for the model.

### 3.2 Blindspot-Aware Joint Optimization

To effectively consolidate the spatial cues learned from under-attended regions, we introduce Blindspot-Aware Joint Optimization (BJO), a bidirectional learning strategy that integrates blindspot feature alignment into the pre-training objective. This optimization encourages the model to learn more discriminative feature representations from blindspot regions while preserving its task-specific learning capabilities and improving its generalization.

**Blindspot feature alignment.** The dynamically selected blindspot token $T^b$ and the standard token $T^s$ are both input into a shared-weight model, producing two distinct levels of point cloud latent representations $F^b$ and $F^s$. Unlike the blindspot input, the complete point cloud retains all original information. Although the masking strategy results in the loss of some regional details, both representations still correspond to the same underlying point cloud entity. To encourage this consistency, we introduce a blindspot feature alignment objective that explicitly aligns global features from the masked and standard inputs. This promotes invariance to specific localized patterns and pushes the model to rely more on global context rather than overfitting to salient regions. The blindspot-aware contrastive loss is defined as:

$$\mathcal{L}_{contra} = -\frac{1}{2a}\sum_i(\log\frac{\exp(H_i^b \cdot H_i^s/\tau_{sim})}{\sum_j \exp(H_i^b \cdot H_j^s/\tau_{sim})} + \log\frac{\exp(H_i^s \cdot H_i^b/\tau_{sim})}{\sum_j \exp(H_i^s \cdot H_j^b/\tau_{sim})}), \tag{6}$$

where $a$ is the number of point clouds in a batch; $\tau_{sim}$ is a temperature hyperparameter; $H_i^b$ and $H_i^s$ are the normalized projection features of $F_i^b$ and $F_i^s$. By omitting the high-attention regions in the masked point clouds, the contrastive objective incentivizes the model to focus on and extract valuable information from less emphasized areas. This process facilitates the learning of a more holistic latent representation, thereby improving the model's capacity to effectively differentiate between various point cloud objects.

**Contrastive learning enhancement.** While contrastive learning is effective at distinguishing features, directly integrating it into existing models can lead to multi-task conflicts. To address this, we propose a phased weighted combination strategy. In the early stage of training, the model focus solely on the original loss to achieve strong performance on the primary task. In the subsequent phase, blindspot feature alignment is gradually introduced through a weighted loss combination, enhancing its ability to recognize global invariant features, while the original task continues to guide the learning process. The proposed total loss is formulated as follows:

$$\mathcal{L}_{total} = \mathcal{L}_{origin} + \lambda\mathcal{L}_{contra}, \tag{7}$$

where $\mathcal{L}_{origin}$ represents the original loss in the existing framework; $\lambda$ is a weight hyperparameter that controls the contribution of blindspot-aware contrastive learning loss. During the pre-training

Table 1: **Robustness on object classification.** We report the classification accuracy (%) with four noisy environments: Gaussian noise, rotation, scaling, and droppoint on ScanObjectNN.

| Dataset | Methods | Gaussian Noise | | Rotation | | | Scaling | DropPoint | |
|---|---|---|---|---|---|---|---|---|---|
| | | $\sigma$=0.01 | $\sigma$=0.03 | X[-30 30] | Y[-30 30] | Z[-30 30] | (0.5, 1.5) | 0.2 | 0.6 |
| OBJ-BG | Point-MAE | 77.5 | 47.2 | 72.1 | 87.6 | 72.5 | 86.2 | 87.4 | 84.9 |
| | **+BlindFormer** | 81.8 | 60.6 | 77.3 | 90.5 | 77.3 | 89.3 | 90.7 | 89.7 |
| | ↑ *Improve* | +4.3 | +13.4 | +5.2 | +2.9 | +4.8 | +3.1 | +3.3 | +4.8 |
| | PointGPT-S | 78.6 | 51.5 | 72.3 | 89.3 | 74.0 | 88.3 | 90.7 | 85.0 |
| | **+BlindFormer** | 81.8 | 57.8 | 76.8 | 91.9 | 79.2 | 90.4 | 91.4 | 86.1 |
| | ↑ *Improve* | +3.2 | +6.3 | +4.5 | +2.6 | +5.2 | +2.1 | +0.7 | +1.1 |
| OBJ-ONLY | Point-MAE | 70.9 | 37.0 | 75.4 | 86.7 | 74.9 | 84.0 | 86.6 | 84.5 |
| | **+BlindFormer** | 76.2 | 54.2 | 78.5 | 88.6 | 79.7 | 86.7 | 88.5 | 87.6 |
| | ↑ *Improve* | +5.3 | +17.2 | +3.1 | +1.9 | +4.8 | +2.7 | +1.9 | +3.1 |
| | PointGPT-S | 71.2 | 39.4 | 72.3 | 89.3 | 74.5 | 86.6 | 89.7 | 85.9 |
| | **+BlindFormer** | 73.3 | 41.3 | 79.9 | 92.3 | 81.8 | 90.0 | 91.2 | 87.4 |
| | ↑ *Improve* | +2.1 | +1.9 | +7.6 | +3.0 | +7.3 | +3.4 | +1.5 | +1.5 |
| PB-T50-RS | Point-MAE | 66.3 | 40.6 | 65.9 | 83.9 | 68.8 | 82.6 | 83.2 | 80.0 |
| | **+BlindFormer** | 69.4 | 43.9 | 68.4 | 85.4 | 70.0 | 84.5 | 84.9 | 82.0 |
| | ↑ *Improve* | +3.1 | +3.3 | +2.5 | +1.5 | +1.2 | +1.9 | +1.7 | +2.0 |
| | PointGPT-S | 63.2 | 34.9 | 63.1 | 85.6 | 67.0 | 84.1 | 86.1 | 82.0 |
| | **+BlindFormer** | 66.5 | 38.8 | 63.8 | 87.0 | 68.2 | 85.6 | 86.7 | 83.5 |
| | ↑ *Improve* | +3.3 | +3.9 | +0.7 | +1.4 | +1.2 | +1.5 | +0.6 | +1.5 |

phase, Point-MAE's original pre-training loss $\mathcal{L}_{origin}$ is equivalent to the reconstruction loss $\mathcal{L}_{re}$. For PointGPT, $\mathcal{L}_{origin}$ refers to the generation loss $\mathcal{L}_{ge}$. Therefore, we optimize the joint learning objective with the phased weighted combination strategy, ensuring that the model not only achieves high-quality reconstructions (or generations) but also learns globally consistent feature representations. Through this strategy, BlindFormer exhibits strong potential for adaptability and scalability across a wide range of multi-task learning scenarios, ultimately improving the model's overall performance.

# 4 Experiments

## 4.1 Experimental Setup

**Datasets.** We evaluate BlindFormer on three benchmarks in point cloud analysis. *ScanObjectNN* [48] comprises approximately 15,000 real-world 3D objects from 15 categories derived from indoor RGB-D scans, presenting challenges like background clutter, occlusions, and sensor noise. *ModelNet40* [59] is a synthetic dataset with 12,311 CAD models across 40 categories, split into 9,843 for training and 2,468 for testing. *ShapeNetPart* [63] contains 16,881 models across 16 categories, each annotated with part labels totaling 50 classes, enabling evaluation of fine-grained part segmentation.

**Backbone models.** To evaluate the seamless integration of the proposed method into existing Transformer-based models for point cloud processing, we apply it to different backbone architectures, specifically Point-BERT, Point-MAE and PointGPT-S. Experimental results across various tasks indicate that the method is adaptable and enhances the performance of these Transformer architectures, thereby demonstrating its versatility and practical applicability.

**Experimental details.** Our input point clouds are obtained by sampling 1,024 points from each raw point cloud. Each point cloud is then divided into 64 patches with 32 points each. The BlindFormer model is pre-trained for a total of 600 epochs: the first 300 epochs focus on the original task alone, and the next 300 epochs incorporate both original pre-training and contrastive learning objectives. We use the Adam optimizer with an initial learning rate of 0.001, a weight decay of 0.05, and a batch size of 128. The learning rate is adjusted using a cosine decay schedule. All experiments are implemented using the PyTorch framework and conducted on four NVIDIA V100 GPUs. More training strategy details and training cost analysis are provided in the Appendix 6.2.

## 4.2 Robustness against point cloud perturbations

To assess the robustness of our BlindFormer framework, we conducted experiments on object classification and part segmentation tasks under different noisy environments, including Gaussian noise, rotation, scaling, and point dropout.

Table 2: **Robustness on part segmentation.** We report the mean Intersection over Union (mIoU) for all classes on ShapeNetPart.

| Methods | Gaussian Noise | | Rotation | | | Scaling | DropPoint | |
|---|---|---|---|---|---|---|---|---|
| | $\sigma$=0.03 | $\sigma$=0.05 | X[-30 30] | Y[-30 30] | Z[-30 30] | (0.5, 1.5) | 0.2 | 0.6 |
| Point-MAE | 74.1 | 71.1 | 77.9 | 80.1 | 74.0 | 82.4 | 77.1 | 64.8 |
| **+BlindFormer** | 75.4 | 72.9 | 80.0 | 81.6 | 76.4 | 83.7 | 78.3 | 67.5 |
| ↑ *Improve* | +1.3 | +1.8 | +2.1 | +1.5 | +2.4 | +1.3 | +1.2 | +2.7 |
| PointGPT-S | 71.5 | 68.2 | 75.9 | 79.1 | 74.5 | 80.6 | 76.5 | 65.0 |
| **+BlindFormer** | 72.3 | 69.9 | 79.2 | 81.0 | 77.2 | 82.2 | 77.5 | 66.9 |
| ↑ *Improve* | +0.8 | +1.7 | +3.3 | +1.9 | +2.7 | +1.6 | +1.0 | +1.9 |

Table 3: **Object classification on ScanObjectNN and ModelNet40.** We report the Top-1 classification accuracy (%) of BlindFormer with Point-MAE and PointGPT-S as backbones respectively. On ScanObjectNN, * denotes using simple rotational augmentation for training. On ModelNet40, * denotes the results obtained by voting.

| Methods | Reference | ScanObjectNN | | | ModelNet40 |
|---|---|---|---|---|---|
| | | OBJ-BG | OBJ-ONLY | PB-T50-RS | |
| *Supervised Learning Only* | | | | | |
| PointNet [35] | CVPR 17 | 73.3 | 79.2 | 68.0 | 89.0 |
| PointNet++ [36] | NeurIPS 17 | 82.3 | 84.3 | 77.9 | 90.2 |
| PointCNN [23] | NeurIPS 18 | 86.1 | 85.5 | 78.5 | 91.7 |
| DGCNN [53] | TOG 19 | 82.8 | 86.2 | 78.1 | 92.0 |
| PRANet [8] | TIP 21 | - | - | 81.0 | 92.9 |
| MVTN [15] | ICCV 21 | - | - | 82.8 | 93.8 |
| PointNeXt [39] | NeurIPS 22 | - | - | 87.7 | 92.9 |
| PointMLP [29] | ICLR 22 | - | - | 85.4 | 94.1 |
| RepSurf-U [42] | CVPR 22 | - | - | 84.3 | 93.8 |
| ADS [18] | ICCV 23 | - | - | 87.5 | 94.0 |
| *with Self-Supervised Representation Learning* | | | | | |
| MaskPoint [26] | CVPR 22 | 89.3 | 88.1 | 84.3 | 92.6 |
| Point-M2AE [67] | NeurIPS 22 | 91.2 | 88.8 | 86.4 | 93.4 |
| PointDif [73] | CVPR 24 | 93.3 | 91.9 | 87.6 | - |
| GPM [24] | CVPR 24 | 90.2 | 90.0 | 84.8 | 93.3 |
| PointMamba [25] | NeurIPS 24 | 90.7 | 88.5 | 84.9 | 93.6 |
| PCM [71] | AAAI 25 | - | - | 88.1 | 93.4 |
| Point-BERT [65] | CVPR 22 | 87.4 | 88.1 | 83.1 | 92.7 |
| **+BlindFormer** | - | 89.5 (+2.1) | 88.6 (+0.5) | 84.5 (+1.4) | 93.1 (+0.4) |
| Point-MAE [34] | ECCV 22 | 90.0 | 88.3 | 85.2 | 93.2 |
| **+BlindFormer** | - | 90.9 (+0.9) | 88.8 (+0.5) | 85.4 (+0.2) | 93.7 (+0.5) |
| PointGPT-S [5] | NeurIPS 23 | 91.6 | 90.0 | 86.9 | 93.3 |
| **+BlindFormer** | - | 92.3 (+0.7) | 91.6 (+1.6) | 87.1 (+0.2) | 93.5 (+0.2) |
| Point-MAE* [34] | ECCV 22 | 92.8 | 91.2 | 89.0 | 93.8 |
| **+BlindFormer*** | - | 93.1 (+0.3) | 91.7 (+0.5) | 89.2 (+0.2) | **94.1** (+0.3) |
| PointGPT-S* [5] | NeurIPS 23 | 93.4 | 92.4 | 89.2 | 94.0 |
| **+BlindFormer*** | - | **94.5** (+1.1) | **93.5** (+1.1) | **89.9** (+0.7) | **94.1** (+0.1) |

**Robustness to object classification.** We conduct extensive experiments on the ScanObjectNN dataset across various noise settings in Table 1. Compared to state-of-the-art models, including PointMAE and PointGPT-S, our method consistently achieves superior classification accuracy in the OBJ-BG, OBJ-ONLY and PB-T50-RS settings. For instance, under Gaussian noise perturbation with $\sigma = 0.03$ on OBJ-ONLY, BlindFormer surpasses PointMAE by 17.2%. Additionally, under rotation perturbation around the x-axis, BlindFormer outperforms PointGPT-S by 7.6%. These results suggest that the enhanced robustness of BlindFormer originates from its attentional blindspot mining strategy with a joint learning objective. By encouraging attention to under-attended regions, this strategy fosters a more comprehensive structural understanding, while blindspot-aware contrastive learning further improves feature discrimination. Consequently, BlindFormer significantly enhances the model's resilience to noise and transformations.

Table 4: **Few-shot classification on ModelNet40.** We report the mean accuracy (%) with standard deviation over 10 independent experiments.

| Methods | 5-way | | 10-way | |
|---|---|---|---|---|
| | 10-shot | 20-shot | 10-shot | 20-shot |
| *Supervised Learning Only* | | | | |
| PointNet | 52.0±3.8 | 57.8±4.9 | 46.6±4.3 | 35.2±4.8 |
| PointNet-CrossPoint | 90.9±1.9 | 93.5±4.4 | 84.6±4.7 | 90.2±2.2 |
| DGCNN | 31.6±2.8 | 40.8±4.6 | 19.9±2.1 | 16.9±1.5 |
| DGCNN-CrossPoint | 92.5±3.0 | 94.9±2.1 | 83.6±5.3 | 87.9±4.2 |
| *with Self-Supervised Representation Learning* | | | | |
| MaskPoint | 95.0±3.7 | 97.2±1.7 | 91.4±4.0 | 93.4±3.5 |
| Point-M2AE | 96.8±1.8 | 98.3±1.4 | 92.3±4.5 | 95.0±3.0 |
| Point-BERT | 94.6±3.1 | 96.3±2.7 | 91.0±5.4 | 92.7±5.1 |
| **+BlindFormer** | **95.1±2.6** | **97.4±2.0** | **91.5±4.7** | **93.0±3.1** |
| Point-MAE | 96.3±2.5 | 97.8±1.8 | 92.6±4.1 | 95.0±3.0 |
| **+BlindFormer** | **96.7±2.7** | **98.2±1.6** | **92.8±4.0** | **95.3±3.2** |
| PointGPT-S | 96.8±2.0 | 98.6±1.1 | 92.6±4.6 | 95.2±3.4 |
| **+BlindFormer** | **97.1±2.3** | **98.8±1.3** | **93.0±4.0** | **95.6±3.0** |

Table 5: **Part segmentation performance on the ShapeNetPart dataset.** We report the mean Intersection over Union (mIoU) across instances (Ins.) and classes (Cls.).

| Methods | Ins. mIoU | Cls.mIoU |
|---|---|---|
| *Supervised Learning Only* | | |
| PointNet | 83.7 | 80.4 |
| PointNet++ | 85.1 | 81.9 |
| DGCNN | 85.2 | 82.3 |
| *with Self-Supervised Representation Learning* | | |
| Point-BERT | 85.6 | 84.1 |
| GPM | 85.8 | 84.2 |
| PointMamba | 86.0 | 84.4 |
| Point-BERT | 85.6 | 84.1 |
| **+BlindFormer** | **86.0** | **84.6** |
| ↑ *Improve* | +0.4 | +0.5 |
| Point-MAE | 86.1 | 84.2 |
| **+BlindFormer** | **86.4** | **85.2** |
| ↑ *Improve* | +0.3 | +1.0 |
| PointGPT-S | 86.2 | 84.1 |
| **+BlindFormer** | **86.7** | **84.8** |
| ↑ *Improve* | +0.5 | +0.7 |

**Robustness to part segmentation.** We assess the robustness of the BlindFormer framework for part segmentation under various noise conditions using the ShapeNetPart dataset, The integration of BlindFormer into the backbone network significantly enhances segmentation performance across different perturbation scenarios. Notably, under rotation perturbation, our approach improves the class mIoU of Point-GPT by 3.3%. These findings demonstrate that our contrastive attention learning strategy effectively preserves the model's reliable segmentation capability in noisy environments.

## 4.3 Downstream tasks for 3D understanding

**Standard object classification.** Table 3 compares our proposed BlindFormer method with existing approaches on the ScanObjectNN and ModelNet40 datasets. Our BlindFormer consistently outperforms these state-of-the-art methods. Compared to Point-MAE [34], BlindFormer achieves higher accuracies by +0.9%, +0.5%, and +0.2% on OBJ-BG, OBJ-ONLY, and PB-T50-RS, respectively. Against PointGPT-S [5], BlindFormer attains improvements of +0.7%, +1.6%, and +0.2% on the same splits. Following recent work setting [25], BlindFormer sets new state-of-the-art results, achieving up to 94.5% on OBJ-BG, 93.5% on OBJ-ONLY and 89.9% on PB-T50-RS. On ModelNet40, BlindFormer achieves 93.7% accuracy without voting and 94.1% with voting, surpassing previous methods without adding additional parameters. These results demonstrate that BlindFormer promotes a comprehensive understanding and feature discrimination for point cloud, leading to improved model performance in standard object classification.

**Few-shot object classification.** Compared to previous approaches, BlindFormer consistently achieves higher accuracy on the ModelNet40 dataset under few-shot learning settings, and the results are presented in Table 4. In the 5-way 10-shot task, our method attains an accuracy of 97.1% with a standard deviation of 2.3%, demonstrating superior generalization with limited labeled data.

**Part segmentation.** We evaluate the effectiveness of our BlindFormer on the part segmentation task, as shown in Table 5. BlindFormer achieves superior performance compared to both traditional supervised models and self-supervised methods. Specifically, our method attains an instance mIoU of 86.4% and a class mIoU of 85.2% with Point-MAE. These results confirm the efficacy of BlindFormer in advancing the state-of-the-art in point cloud segmentation.

## 4.4 Ablation Studies

We conduct extensive experiments with Point-MAE on ScanObjectNN under Gaussian noise to validate the effectiveness of each component. Table 6 summarizes the ablation study on different combinations for the OBJ-BG and OBJ-ONLY settings.

Table 6: **Ablation studies of components in BlindFormer.** We report the overall accuracy (%) on ScanObjectNN with Point-MAE. The settings adopted by BlindFormer are  marked .

(a) Blindspot Mining Strategy and Loss Optimization Function.

| Blindspot Mining Strategy | $\mathcal{L}_{origin}$ | $\mathcal{L}_{contra}$ | OBJ-BG | OBJ-ONLY |
|---|---|---|---|---|
| NO Mining | ✓ | - | 47.2 | 37.0 |
| Random Blindspot Mining | ✓ | - | 50.5 | 41.3 |
| Attentional Blindspot Mining | ✓ | - | 56.9 | 50.4 |
| Random Blindspot Mining | - | ✓ | 47.7 | 38.3 |
| Attentional Blindspot Mining | - | ✓ | 53.8 | 47.5 |
| Random Blindspot Mining | ✓ | ✓ | 52.7 | 43.4 |
| **Attentional Blindspot Mining** | ✓ | ✓ | **60.6** | **54.2** |

| (b) Mask Ratio. | | | (c) Probability Temperature. | | | (d) Contrastive Loss Weight. | | |
|---|---|---|---|---|---|---|---|---|
| $\mathcal{R}$ | OBJ-BG | OBJ-ONLY | $\tau_{pro}$ | OBJ-BG | OBJ-ONLY | $\lambda$ | OBJ-BG | OBJ-ONLY |
| 0.2 | 53.1 | 45.6 | 0.3 | 59.2 | 53.1 | 0.4 | 59.8 | 53.5 |
| 0.4 | 57.8 | 52.1 | **0.5** | **60.6** | **54.2** | **0.6** | **60.6** | **54.2** |
| **0.6** | **60.6** | **54.2** | 0.7 | 60.1 | 53.8 | 0.8 | 58.2 | 52.1 |
| 0.8 | 45.9 | 36.8 | 0.9 | 58.7 | 52.9 | 1.0 | 57.5 | 51.8 |

**Blindspot mining strategy.** As shown in Table 6(a), we explore how different blindspot mining strategies affect performance. Without Mining, the baseline model achieves 47.2% (OBJ-BG) and 37.0% (OBJ-ONLY) accuracy. The *Random Blindspot Mining* provides a moderate improvement, indicating that even coarse region perturbations help the model discover underutilized cues. However, the *Attentional Blindspot Mining* yields the best results, achieving 56.9% (OBJ-BG) and 50.4% (OBJ-ONLY) accuracy with the original loss $\mathcal{L}_{origin}$. Similar trends can also be observed when maintaining the same loss optimization function. By masking out high-attention regions, this strategy explicitly forces the model to improve comprehensive understanding and robustness by learning from less salient areas and enhancing its generalization capabilities.

**Loss optimization function.** We also analyze the effect of blindspot-aware joint learning in Table 6(a). Compared to the baseline, applying contrastive learning alone with attentional blindspot mining already yields notable improvements (+6.6% on OBJ-BG and +10.5% on OBJ-ONLY). However, contrastive loss alone underperforms relative to the original task loss with attentional mining strategy. This is because $\mathcal{L}_{contra}$ only focuses on aligning blindspot features across masked views, but lacks the fine-grained semantic supervision that $\mathcal{L}_{origin}$ provides for accurate reconstruction or generation. It highlights the role of contrastive loss as a regularizer rather than a substitute. When both objectives are jointly optimized, we observe the best performance (60.6% on OBJ-BG and 54.2% on OBJ-ONLY). The original loss ensures task-oriented fidelity, while the contrastive objective regularizes the learning process, further encouraging the model to extract robust and invariant features across diverse blindspot regions. This synergy confirms that Blindspot-Aware Joint Optimization effectively leverages both local precision and global consistency, resulting in superior representation learning.

**Effect of hyperparameters.** We further explore the effects of varying hyperparameter in BlindFormer. Regarding the masking ratio in Table 6(b), the model achieves optimal performance at $\mathcal{R} = 0.6$, yielding classification accuracies of 60.6% on OBJ-BG and 54.2% on OBJ-ONLY. However, a higher mask ratio ($\mathcal{R} = 0.8$) negatively impacts performance due to excessive information loss, highlighting the need for a balance between data complexity and sufficient feature retention. As shown in Table 6(c) shows the effect of the temperature $\tau_{pro}$ in dynamic masking. Setting $\tau_{pro} = 0.5$ achieves the best accuracy, as it enables selective masking of high-attention regions while preserving dynamic adaptability. Similarly, the contrastive loss weight in Table 6(d) is crucial, with $\lambda = 0.6$ providing the best trade-off between the original and contrastive loss components. This optimal weighting enhances feature discrimination and generalization while preserving task-specific accuracy.

# 5 Conclusion

In this work, we present BlindFormer, a robust self-supervised framework for point cloud understanding. By introducing an attentional blindspot mining strategy, BlindFormer identifies and dynamically masks high-attention regions, encouraging the model to explore under-attended and informative

areas. Building upon this, the blindspot-aware joint optimization scheme effectively combines the strengths of task-specific and contrastive objectives to further enhance blindspot representation learning. Extensive experiments on challenging benchmarks validate the effectiveness of our approach, including object classification, few-shot learning, and part segmentation. These experimental results demonstrate that BlindFormer not only improves performance under standard conditions, but also exhibits strong resistance to perturbations. Our findings point toward the importance of attention regularization in 3D learning and provide a foundation for more resilient and adaptive models.

## Acknowledgments

This work was supported in part by the Joint Funds of the National Natural Science Foundation of China under Grant U24A20248. The work described in this paper was supported in part by the Research Grants Council of the Hong Kong Special Administrative Region, China, under Project T45-401/22-N and under Project CUHK 14200824.

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

# 6 Appendix

## 6.1 Preliminary

**Transformer-based self-supervised learning.** Given a point cloud $X \in \mathbb{R}^{p \times 3}$, we utilize Farthest Point Sampling (FPS) and K-Nearest Neighbors (KNN) algorithms to identify $n$ center points $C$ and their corresponding $k$ nearest neighbors, forming $n$ point patches $P$. Following the previous methods [34, 5], each point patch is normalized to integrate local information. A lightweight token embedding module, implemented via PointNet, subsequently transforms these normalized local patches into trainable point tokens $T$. These point tokens, together with positional embeddings, are input into the transformer blocks to produce latent representations $F$. For different tasks, these latent representations are input into task-specific heads, where they are transformed into specific representations adapted to the task. The learning pipeline based on the Transformer architecture is as follows:

$$F = Transformer(T), \tag{8}$$
$$R = Head_{Task-Spec.}(F). \tag{9}$$

For Point-MAE, $Head_{Task-Spec.}$ denotes the reconstruction head. For PointGPT, $Head_{Task-Spec.}$ denotes the prediction head.

## 6.2 Training Strategy with Cost Analysis

Given that BlindFormer determines mask patches based on the attention weights of the backbone network, we suggest two strategies for obtaining these attention weights. The first strategy initializes the network with random attention and applies the attentional blindspot mining for adaptive attention refinement during subsequent training. Following standard protocol, the model undergoes pre-training for 300 epochs. This approach does not incur any additional training overhead. The second strategy, by contrast, employs attention learned from the standard branch for initialization, aiming to dynamically adjust the model's dependencies in a targeted manner. This method necessitates 300 epochs of pre-training in the standard branch, followed by another 300 epochs in the dual branch, resulting in a total of 600 epochs.

Furthermore, we introduce BlindFormer during the fine-tuning phase of downstream tasks to further evaluate the scalability and effectiveness of our approach. Two strategies are employed here as well: one leverages the pre-trained attention for initialization, while the other requires an additional 300 epochs of training to obtain attention learned from the standard branch for initialization. It should be noted that the dual-branch training framework results in a twofold increase in the data processed per batch. Consequently, we introduce a comparative baseline with a doubled batch size to ensure a fair evaluation.

Table 7: **Training cost analysis.** We report the classification accuracy (%) on ScanObjectNN.

| DataSet | Methods | Pre-Training Epoch | | Finetune Epoch | |
|---|---|---|---|---|---|
| | | 300 | 600 | 300 | 600 |
| OBJ-BG | Point-MAE | 90.0 | 90.2 | 90.0 | 90.2 |
| | **+BlindFormer** | 90.5 | 90.5 | 90.5 | 90.9 |
| | ↑ *Improve* | +0.5 | +0.3 | +0.5 | +0.7 |
| | PointGPT-S | 91.6 | 91.7 | 91.6 | 91.9 |
| | **+BlindFormer** | 91.9 | 91.9 | 92.1 | 92.3 |
| | ↑ *Improve* | +0.3 | +0.2 | +0.5 | +0.4 |
| OBJ-ONLY | Point-MAE | 88.3 | 88.5 | 88.3 | 88.5 |
| | **+BlindFormer** | 88.8 | 89.8 | 89.2 | 88.8 |
| | ↑ *Improve* | +0.5 | +1.3 | +0.9 | +0.3 |
| | PointGPT-S | 90.0 | 90.2 | 90.2 | 90.2 |
| | **+BlindFormer** | 90.5 | 91.4 | 90.9 | 91.6 |
| | ↑ *Improve* | +0.5 | +1.2 | +0.7 | +1.4 |

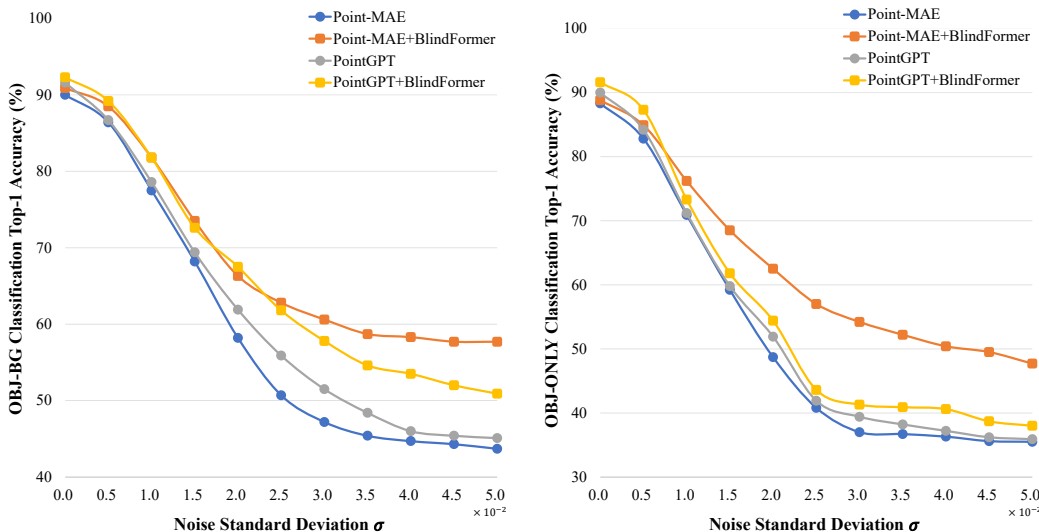

Figure 3: **Gaussian noise analysis on ScanObjectNN.** While the performance of existing methods decmidrules sharply with increasing Gaussian noise, this issue is mitigated by incorporating Blind-Former. Notably, when Point-MAE is used as the backbone network, our BlindFormer significantly enhances its robustness, resulting in minimal accuracy degradation.

Our experimental results, presented in Table 7, demonstrate the inherent advantages of our Blind-Former over existing approaches (such as Point-MAE and PointGPT-S) under the same training time, training batch size and training phase. For Point-MAE, with 300 pre-training epochs or 300 fine-tuning epochs, BlindFormer achieves an accuracy of 90.5% on the OBJ-BG dataset, surpassing Point-MAE's 90.0% by a margin of 0.5%. This improvement persists when both methods are trained for 600 epochs during the fine-tuning phase, with BlindFormer reaching 90.9% accuracy compared to Point-MAE's 90.2%. Similarly, when evaluating against PointGPT-S, BlindFormer continues to exhibit superior performance. With both models trained for 300 fine-tuning epochs on OBJ-ONLY, BlindFormer attains an accuracy of 90.9% compared to PointGPT-S's 90.2%. Even when the training epochs are extended to 600, BlindFormer maintains its advantage, achieving 91.6% accuracy, outperforming PointGPT-S by 1.4%. On the OBJ-BG dataset, a similar pattern is observed, where BlindFormer consistently outperforms PointGPT-S regardless of training duration.

The superior performance of BlindFormer across various datasets, training epochs, and application phases validates the efficacy of our framework. It demonstrates the performance gains of BlindFormer are not a consequence of longer training times but are a direct result of designed framework contributions—namely, the attentional blindspot mining strategy with blindspot-aware joint optimization. By focusing on under-attended regions and enhancing feature discrimination, BlindFormer effectively captures both global and local features, leading to enhanced robustness and generalization.

## 6.3 Robutness Analysis

**Robustness to noise corruptions.** We further evaluate the robustness of our method against existing approaches under Gaussian noise conditions using the OBJ-BG and OBJ-ONLY subsets of the ScanObjectNN dataset. To simulate noisy point clouds, we add Gaussian noise $X \sim \mathcal{N}(0, \sigma^2)$ to all points, incrementally increasing the noise level by varying $\sigma$ from 0 to 0.05 with step size = 0.005. As illustrated in Figure 3, while the accuracy of all methods decmidrules as the noise standard deviation $\sigma$ increases, BlindFormer exhibits a slower performance degradation, demonstrating its superior ability to handle noisy point clouds. Notably, BlindFormer significantly improves the robustness of the Point-MAE backbone and outperforms baseline methods such as Point-MAE and PointGPT, particularly under extreme noise conditions ($\sigma = 0.05$). This improvement can be attributed to our attentional blindspot mining strategy, which encourages the model to focus on under-attended regions, thereby enhancing its capacity to capture comprehensive global structural information from point clouds. By not solely relying on salient local features, BlindFormer mitigates sensitivity to

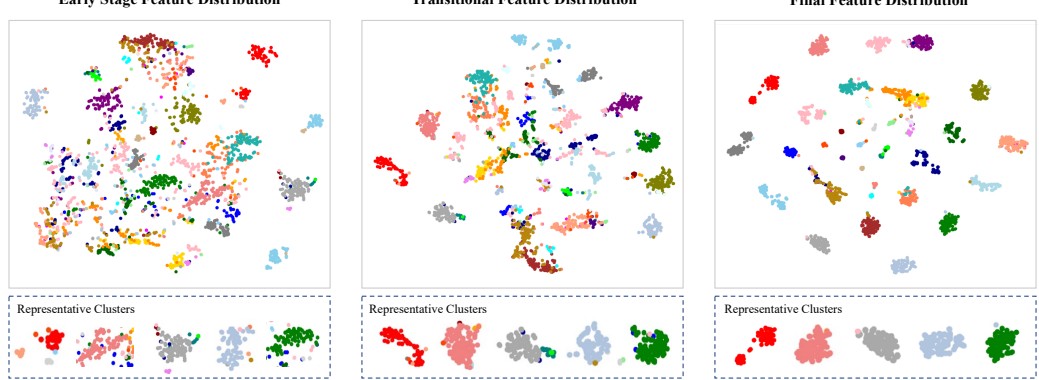

Figure 4: **Feature distribution visualization on ModelNet40. Top:** An overview of the evolution of feature distributions across all 40 classes. **Bottom:** Detailed depiction of the evolution of feature distributions for selected typical classes.

noise-induced perturbations. Additionally, the integration of contrastive learning with the original task further refines feature discrimination, enabling the model to distinguish subtle variations in data even under noisy conditions. The consistently strong performance across both the OBJ-BG and OBJ-ONLY datasets underscores the versatility and reliability of BlindFormer in diverse settings.

**Robustness to local corruptions.** Following the standard setting[21, 43, 51], we conduct additional experiments to evaluate the robustness of BlindFormer against local corruptions, as shown in Table 8. We consider two types of local corruptions: (1) LocalDrop, which drops $C$ local clusters, and (2) LocalAdd, which adds $C$ local clusters. Each cluster consists of $K$ nearest points from a randomly selected cluster center point. We used $K = 100$ in both cases.

By integrating BlindFormer, both Point-MAE and PointGPT-S demonstrate significant performance improvements under local corruptions. Specifically, under severe interference ($C$=9), BlindFormer enhances Point-MAE by 5.2% for LocalDrop and 5.3% for LocalAdd, while improving PointGPT-S by 4.1% for LocalDrop and 6.3% for LocalAdd. We attribute this improvement to BlindFormer's ability to activate global structural attention, which not only enhances feature discrimination but also enables the model to learn more robust representations against local perturbations.

Table 8: **Robustness to local corruptions on ScanObjectNN.**

| Methods | LocalDrop | | | LocalAdd | | |
|---|---|---|---|---|---|---|
| | $C$=3 | $C$=6 | $C$=9 | $C$=3 | $C$=6 | $C$=9 |
| Point-MAE | 87.9 | 83.6 | 82.1 | 83.5 | 81.4 | 77.3 |
| **+BlindFormer** | 90.4 | 88.7 | 87.3 | 88.8 | 86.4 | 82.6 |
| ↑ *Improve* | +2.5 | +5.1 | +5.2 | +5.3 | +5.0 | +5.3 |
| PointGPT-S | 88.5 | 86.6 | 84.0 | 87.7 | 84.5 | 79.9 |
| **+BlindFormer** | 91.7 | 89.8 | 88.1 | 90.7 | 89.0 | 86.2 |
| ↑ *Improve* | +3.2 | +3.2 | +4.1 | +3.0 | +4.5 | +6.3 |

In real-world applications, 3D data is often affected by noise from sensor inaccuracies and environmental factors, making BlindFormer's robustness to various corruptions especially valuable. The performance under such conditions demonstrates its practicality for tasks where data quality is uncertain, underscoring the effectiveness of BlindFormer and its advantage over existing Transformer-based methods.

## 6.4 Feature Distribution Evaluation

Figure 4 illustrates the evolution of the feature distribution using t-SNE during the fine-tuning of BlindFormer, with Point-MAE as the backbone, on the ModelNet40 dataset. In the early stage feature distribution, the feature space is highly scattered with overlapping clusters, indicating that the backbone has not yet learned to effectively discriminate between different classes. As the backbone starts to align global representations from standard branch and blindspot branch based on attention-driven blindspot mining, the transitional feature distribution shows a notable improvement, with clusters becoming more distinct. However, there still remains some inter-class overlap.

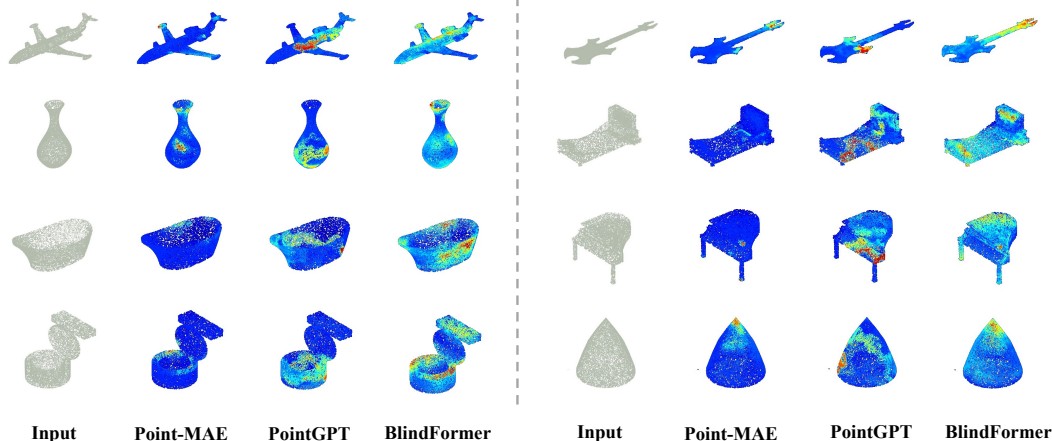

| Input | Point-MAE | PointGPT | BlindFormer | Input | Point-MAE | PointGPT | BlindFormer |

Figure 5: **Attention visualization of BlindFormer with Point-MAE and PointGPT.** Patches with high attention are closer to red, while patches with low attention are closer to blue. Point-MAE is employed as the backbone of our proposed BlindFormer.

In the final feature distribution, the clusters are well-separated and compact, reflecting a highly discriminative feature space. The backbone has successfully learned to distinguish between different classes with a high degree of accuracy. The representative clusters at the bottom of each visualization further emphasize this progression, showing a clear transition from mixed and overlapping clusters in the early stages to well-defined and isolated clusters in the final stage. These visualizations highlight the effectiveness of the BlindFormer, demonstrating a clear trajectory of improvement in feature discrimination, culminating in a robust and well-defined feature space.

## 6.5    Qualitative Analysis

As shown in Figure 5, we visualize the classification heatmaps generated by Point-MAE, PointGPT, and our proposed BlindFormer, revealing significant distinctions in how these model attend to various regions of the point clouds. BlindFormer exhibits a more balanced and comprehensive activation across both prominent and under-represented areas of the input data. Notably, on 2,468 ModelNet samples, Point-MAE exhibits a highly concentrated attention distribution, with top 10% of patches capturing 61.2% of total attention weights. This excessive reliance on a small subset of high-attention regions limits the model's ability to generalize and adapt to perturbations. In contrast, BlindFormer demonstrates a more balanced attention distribution, requiring 36% of patches to reach the same cumulative attention weight. This redistribution indicates that BlindFormer effectively engages under-attended regions, reinforcing its ability to capture global structural information.

The observation aligns with our motivation in the introduction—existing Transformer-based models tend to prioritize a limited set of salient regions, potentially neglecting latent information crucial for robust feature learning. By integrating our attentional blindspot mining strategy, BlindFormer compels the model to explore less prominent areas, leading to more comprehensive feature extraction. Furthermore, blindspot-aware contrastive learning refines feature discrimination, ensuring a more stable and generalized understanding of the point cloud. The richer and more evenly distributed activations in BlindFormer's heatmaps further substantiate its ability to mitigate the limitations of prior methods, improving both robustness and generalization in 3D understanding.

## 6.6    Domain Adaptation Discussion

Although our work does not explicitly target domain adaptation, we find that BlindFormer improves generalization under input perturbations, which is conceptually aligned with the goals of domain generalization. For example, under Gaussian noise, categories with weaker original attention show significant accuracy gains (e.g., "display" from 38.10% to 76.19%, "sofa" from 54.76% to 80.95%). Additionally, BlindFormer better distinguishes between morphologically similar categories (e.g., average accuracy of "desk" and "table" improves from 50.19% to 74.26%), demonstrating its robustness against subtle inter-class ambiguities.

These preliminary observations suggest that BlindFormer enables the model to handle challenging or less typical samples more effectively. This behavior naturally enhances resistance to distribution shifts, as it reduces dependence on domain-specific or spurious correlations. We believe such robustness makes BlindFormer a promising candidate for future exploration in out-of-distribution and domain-adaptive settings.

## 6.7 Limatation Analysis

Despite the significant improvements achieved by BlindFormer, there are still areas that offer opportunities for further enhancement. For example, while our method has been validated on specific datasets, applying it to a broader range of datasets could further demonstrate its generalizability and robustness. Additionally, although we have shown that BlindFormer integrates seamlessly with certain Transformer-based architectures, exploring its compatibility with an even wider variety of models could highlight its versatility even more. These considerations open avenues for future research to build upon our work and continue advancing the field of point cloud analysis.

## 6.8 Future Works

While the proposed BlindFormer framework has shown significant improvements in point cloud analysis tasks, there are several promising directions for future research to further enhance its capabilities and applications.

One potential avenue is the integration of multi-modal data sources to enrich point cloud representations. By incorporating complementary information from modalities such as images, textual descriptions, or LiDAR intensity values, the model can leverage cross-modal correlations to learn more comprehensive and robust feature embeddings. This multi-modal fusion could enhance the model's ability to understand complex scenes and improve performance in several tasks. Another direction is the exploration of hierarchical or multi-scale feature learning within the BlindFormer framework. By capturing features at various spatial resolutions, the model can better represent both local geometric details and global structural contexts. This enhancement could be particularly beneficial for handling large-scale point clouds or scenes with significant variations in point densities. Additionally, optimizing the computational efficiency of the attentional blindspot mining and blindspot-aware joint optimization is an important consideration for real-world applications. Investigating lightweight architectures or efficient training strategies could make the model more suitable for deployment in resource-constrained environments, such as mobile robots or embedded systems used in autonomous driving. Lastly, applying the BlindFormer approach to other types of data representations, such as meshes or voxels, could broaden its applicability across different domains in 3D data processing. Exploring transfer learning techniques between these representations may also provide insights into shared structures and features among various 3D data forms.

By pursuing these future research directions, we aim to further advance the capabilities of Blind-Former, contributing to the development of more robust, efficient, and versatile models for point cloud analysis. These enhancements have the potential to impact a wide range of applications, including robotics, augmented reality, virtual reality, and autonomous navigation, by enabling more accurate and comprehensive understanding of complex 3D environments.

