# OpenReview forum: "What We Miss Matters: Learning from the Overlooked in Point Cloud Transformers"
_NeurIPS.cc/2025/Conference — NeurIPS 2025 poster_

### Official Review · Reviewer_mAUX · 2025-06-01

**Clarity:** 4
**Significance:** 3
**Originality:** 3
**Rating:** 4
**Confidence:** 4

**Summary:**

This manuscript proposes a novel contrastive learning framework for object-level point cloud transformers, where the core idea is to suppress the salient attention regions, forcing the model to expand its perceptual field and therefore increasing the robustness. Specifically, the patches with high attention scores in the last transformer layer during the first forward pass are occluded during the second pass. Results of the two passes are forced to be similar through contrastive learning. Experiments demonstrate consistent improvements on 3 different backbones.

**Questions:**

1.	A fair baseline: As discussed in weakness 2, your framework doubles the training computation per data sample, which is equivalent to doubling the batch size. IMHO the proper baselines are the baseline models with twice the batch size. It is important to separate your contribution apart from the increment that is brought by simply using a greater batch size, possibly through some experiments.

2.	Additional comparison methods: How does your framework compare with other occlusion-oriented pretraining methods, such as MSC [2] and Sonata[3]? Note that this question is optional and omitting this question does not negatively affect my final scores.

3.	Cost analysis: Can you provide detailed VRAM and time analysis compared to baselines?

4.	Design choice questionable: As discussed in weakness 3, the claims from L143 to L147 is not well supported. Can you make a mathematical derivation of the properties of your dynamic blindspot generation strategy?

Overall, the proposed method is original and novel, but not without problems such as cost analysis, fairness, and design choices. The strengths outweigh the weaknesses, which grants a borderline accept. Although I do hope this paper can be made better through drawing experiences of other fields (weakness 4), this will be out of scope for this paper. I would keep the right to adjust my rating according to the author response.

[2] Xiaoyang Wu, Xin Wen, Xihui Liu, and Hengshuang Zhao. Masked scene contrast: A scalable framework for unsupervised 3d representation learning. In CVPR, 2023.

[3] Xiaoyang Wu, Daniel DeTone, Duncan Frost, Tianwei Shen, Chris Xie, Nan Yang, Jakob Engel, Richard Newcombe, Hengshuang Zhao, and Julian Straub. Sonata: Self-supervised learning of reliable point representations. arXiv preprint arXiv:2503.16429, 2025 (Accepted in CVPR 25)

**Ethical Concerns:**

["NO or VERY MINOR ethics concerns only"]

**Final Justification:**

My major concerns have been adequately solved in the discussion period. While Q1 remains, it is minor and does not affect the final rating, and I have suggested the authors include such experiments in the revision. After checking with other reviewers' opinions, I would remain the current rating on this paper (4, borderline accept).

**Limitations:**

yes

**Paper Formatting Concerns:**

1. Eq. 5 should be formulated with union subtraction operation or complement operation, i.e., seeing $P$ as a set of patch features, and this equation subtracts (or takes the complement of) the masked patches $P_{mask}$ from the full set $P$.

2. Eq. 6 $H^m_i$ should be $H^b_i$.

**Quality:**

3

**Strengths And Weaknesses:**

Strengths:

1.	The paper is clearly written and well structured. The core idea is exceptionally novel and provides valuable insight to existing methods.

2.	The core finding is very valuable. The authors discover that over-reliance on several high attention patches affects model robustness. This has highlighted the importance of analyzing attention score distribution on point cloud transformers, which is not very popular in the point cloud transformer community to the best of my knowledge, compared to the wide adoption of such analysis in VLM community.

3.	The findings are solid and well backed up by ablation and extensive experiments on various benchmarks.

Weaknesses:

1.	Cost analysis is almost absent. From my understanding, the pipeline doubles the training computation cost compared to baseline models due to the two-pass training pipeline. When adopting the learned attention initialization strategy (Appendix L524), a complete additional training is also required. While the authors provide the performance superiority in the so-called cost analysis of Tab. 7, no actual VRAM costs or time costs are mentioned.

2.	Due to the same reason as W1, the baselines are not completely fair. Your framework doubles the training computation per data sample, which is equivalent to doubling the batch size. A fair baseline with double the batch size is needed. See question 1 for details.

3.	Eq. 3 design is weired (this is not a major weakness). Eq. 3 simplifies to $p_{dy}=log(-softmax(S/\tau) / log(\epsilon))$, where $\epsilon \sim U[0,1]$. This means that every softmax significance score with range $(0, 1)$ is randomly scaled by a factor $ -1/log(\epsilon)$ which is of range $(0,+\infty)$. Therefore, the effect of noise is significantly higher than original softmax significance scores. I do not know whether such high level of randomness is vital for your design, does this mean that your attention-guided masking strategy is actually similar to random masking?

4.	Designs can be drawn from other communities (this is not a major weakness, more like a piece of advice). In the ViT community, such occasional bloated attention scores are called attention artifacts, and their effect has been determined as the rendezvous point to aggregate global information, since their local geometry is the least important and can be charged for other purposes with little loss. All tokens will have high attention scores with these tokens. Furthermore, adding a `register token’ [1] can restore the geometrical discriminativeness of attention maps. Maybe their experience can be applied to solve your problem.

[1] Timothée Darcet, Maxime Oquab, Julien Mairal, and Piotr Bojanowski. Vision transformers need registers. arXiv preprint arXiv:2309.16588, 2023.

---

> ### Author Rebuttal · Authors · 2025-07-31
>
> We thank the reviewer for the positive evaluation and the valuable feedback. We believe that we have fully addressed your concerns and questions, and will incorporate all points mentioned below in the final version. We would be happy to address any further questions you might have.
> ***
> > **Weakness** 1 & **Question** 3: Detailed cost analysis.
>
> We appreciate the reviewer’s concerns regarding computational cost and have conducted a more detailed analysis.
>
> (1) Learned Attention Initialization. Acquiring learned attention indeed requires additional training (Appendix Line 524). However, for fairness, we apply the same training configuration to the baseline. The 600-epoch results in Table 7 all use learned attention initialization (The first 300 rounds are used for initial attention learning). Regardless of whether the attention initialization is random or learned, BlindFormer ``consistently outperforms the baseline`` under identical training schedules, demonstrating that the gains stem from blindspot mining rather than longer training.
>
> (2) VRAM usage. Using gpustat to monitor GPU memory at the same batch size (bs=32), BlindFormer occupies 11,119 MB, compared to 11,043 MB for Point-MAE, a marginal increase of about 0.7%. This includes ``all runtime allocations`` (model parameters, optimizer states, activation buffers, and CUDA memory fragmentation), not just model weights.
>
> (3) Training time. Due to the dual-branch design, BlindFormer incurs approximately 2× training overhead. For instance, on OBJ-BG classification training, one epoch takes 43s versus 22s for Point-MAE. Crucially, this additional cost is confined to training and ``does not affect inference speed``, making the method practical for deployment. For a fairer comparison, we explain this in detail in the next weakness.
> ***
> > **Weakness** 2 & **Question** 1: Fair comparison of data samples.
>
> We thank the reviewer for raising this concern. To further clarify, we conduct additional experiments on Point-MAE and PointGPT-S with double batch size under the same training schedule, as shown in Table A. The results on OBJ-BG show only marginal gains (+0.2% on Point-MAE and +0.3% on PointGPT-S), below the improvements obtained by BlindFormer (+0.9% and +0.7%, respectively). Similar results are observed on OBJ-ONLY. This confirms that BlindFormer’s performance boost arises from its blindspot mining and feature alignment mechanisms, rather than increased batch size. Moreover, BlindFormer’s inference cost remains identical to the baseline, as the blindspot branch is only used in training.
>
> Table A. Performance with the same data sample size.
> | Method | OBJ-BG | OBJ-ONLY |
> | ---- | ---- | ---- |
> | Point-MAE w/o 2x data | 90.0 | 88.3 |
> | Point-MAE w/ 2x data | 90.2 | 88.5 |
> | +BlindFormer | 90.9 | 88.8 |
> | PointGPT-S w/o 2x data | 91.6 | 90.0 |
> | PointGPT-S w/ 2x data | 91.9 | 90.2 |
> | +BlindFormer | 92.3 | 91.6 |
> ***
> > **Weakness** 3 & **Question** 4: Design choice of perturbation probability.
>
> We appreciate your simplification of our formula and would like to clarify the role of the perturbation term.
>
> (1) Derivation of probability distribution. Your series of derivations are reasonable. Please allow us to continue the derivation from the perspective of probability. For a random number $\epsilon \sim U[0, 1]$, the cumulative distribution function $F_Y(y)=P(-log\epsilon \le y) = P(\epsilon \ge e^{-y}) = 1- P(\epsilon \le e^{-y}) = 1- e^{-y}$. In other words, although the range of $-log\epsilon$ is theoretically $(0, +\infty)$, ``it is highly concentrated``. (e.g. $P(-log\epsilon \le 3)\approx0.95$, $P(-log\epsilon \le 7)\approx0.999$). In practice, the attention score is still the dominant variable, and the perturbation only slightly alters mask selection to promote diversity.
>
> (2) Additional theoretical explanation. The perturbation term $-log(-log\epsilon)$ is called ``Gumbel noise``, which is a random variable sampled from the Gumbel distribution. It is widely used in reinforcement learning and generative models to introduce randomness. Given our core contribution is not an improvement based on this, more detailed mathematical theory and related work can be referred to [1, 2]. Thank you again for your concern. We will add relevant literature to the final version to facilitate readers' understanding.
>
> [1] Maddison, et al. "The Concrete Distribution: A Continuous Relaxation of Discrete Random Variables." ICLR, 2017.
>
> [2] Oberst, Michael, and David Sontag. "Counterfactual off-policy evaluation with gumbel-max structural causal models." ICML, 2019.
> ***
> > **Weakness** 4: Design inspiration from the ViT community.
>
> We thank the reviewer for their valuable suggestions. As you mentioned, in the ViT community, adding registration markers has been shown to mitigate the loss of geometric discriminability. BlindFormer approaches this issue from a different perspective, encouraging the model to explore blindspots in sparse point clouds. Ultimately, they both achieve the common goal of providing smoother feature maps and attention maps for downstream visual processing. We agree that integrating these ideas will further improve our framework. We will consider studying registration markers as a promising extension in future work.
> ***
> > **Question** 2: Additional comparison methods.
>
> We thank the reviewer for pointing out related occlusion-oriented pretraining works. MSC and Sonata focus on improving pretraining efficiency and mitigating the geometric shortcut problem, respectively. Both methods combine reconstruction and contrastive objectives, demonstrating excellent performance in 3D scene understanding. BlindFormer specifically addresses attentional blindspots by dynamically reallocating attention to under-explored regions, improving robustness to occlusion and sparsity. We view our blindspot mining as complementary rather than competitive and will include a discussion of MSC and Sonata to the related work section in the final revision.
> ***
> > **Paper Formatting Concerns**: Formula optimization.
>
> Thank you for your suggestions for revisions to the formulas and symbols. We have re-reviewed the entire paper and corrected some minor writing errors to ensure clarity and accuracy in the final version.

---

> > ### Comment · Reviewer_mAUX · 2025-08-03
> >
> > Thank you for the detailed response. All my concerns have been adequately solved as following.
> >
> > (1) Efficiency: The authors have demonstrated marginal additional VRAM usage (+0.7%) and confirmed that the method doubles the training time yet keeps inference time unchanged. The authors are suggested to further incorporate a tabular comparison in the revision to further enhance clarity.
> >
> > (2) Baseline fairness: While it is true that doubling the batch size increases the baseline performance slightly, the increment is smaller than that of the proposed method. Therefore the method's raw increment is slightly cut but nonetheless still significant. The authors should include such baselines in the main table and include discussions in the revision for comparison fairness.
> >
> > (3) Design choice of Eq. 3: Thanks for the clarification, the design choice is now clear and reasonable to me.

---

> > > ### Author Response · Authors · 2025-08-04
> > > **Thanks for suggestions**
> > >
> > > We sincerely thank the reviewer for reading and responding to our rebuttal. We have carefully incorporated your suggestions and will further enhance the paper's clarity by expanding our discussion of efficiency costs and fair baseline in the final version. Thank you again for your valuable insights that have strengthened our paper.
> > >
> > > We hope our detailed responses have addressed your concerns satisfactorily and would be grateful if you could re-evaluate our paper based on the revisions. We are happy to address any further concerns/questions you might have.

---

### Official Review · Reviewer_oJo8 · 2025-06-20

**Clarity:** 3
**Significance:** 2
**Originality:** 3
**Rating:** 4
**Confidence:** 3

**Summary:**

The paper introduces BlindFormer, a masking strategy for blind spots in 3D point clouds to enhance the focus on non-salient regions, thus improve generalization after self-supervised pre-training. A self supervised auxiliary objective induces the model to focus on regions of the point cloud that would be otherwise ignored. A wide range of experiments are conducted to show its performance.

**Questions:**

1. Authors state that over-attending to certain areas of the point cloud make transformers more sensitive to noise or missing points in that area. But are we sure that they would still attend these regions if noise or missing points occurred? This is considering that attention uses all the information around instead of focusing on points independently.

2. In Eq. 3, what happens if the mask corresponds only to the strength of the attention scores? Is the Uniform component really improving the diversity of the masking?

3. What is H^m_i in Eq. 6? Is it an intermediate projection to contrast H^s_i? If so, this loss approximate InfoNCE.

4. From lines 183 to 189, the authors mention that in the early stages of training, the model only focuses on the main task (original loss function), while the subsequent phase focuses on the blindspot feature alignment. This means that the term \lambda in Eq. 7 will be modulated during training? This is not clear in the paper. An explanation comes in the experimental details, but then the \lambda will not be used?

**Ethical Concerns:**

["NO or VERY MINOR ethics concerns only"]

**Final Justification:**

The authors replied to my concerns, and the paper proposes a interesting perspective for the pre-training of 3D models. I increase my score to borderline accept, hoping to be put into consideration in contrast with the other reviewers' final scores.

**Limitations:**

Yes

**Paper Formatting Concerns:**

Formatting is correct.

**Quality:**

3

**Strengths And Weaknesses:**

Strengths:

1. The proposed ideas are interesting and correlate well with the motivations.
2. BlindFormer is evaluated on two main tasks, and a few other downstream tasks. This, combined with using different datasets and pre-training strategies, makes the experimental part of the paper sound and rigorous.

Weaknesses:

1. The motivation is interesting, but there is a sense of lack of evidence or sustain that the negative effects of not attending to ignored regions really affect.
2. The empirical results show small improvements in object classification, specially with respect to supervised alternatives (Table 3 w.r.t. ADS). The same happens against standard SSL methods in Tables 4 and 5, considering that standard deviation is relatively large.This issue puts the significance of the method into question.
4. Minor writing errors need to be resolved. For example, line 193 "we optimizes".

---

> ### Author Rebuttal · Authors · 2025-07-31
>
> We thank the reviewer for the time and the valuable feedback. We believe that we have fully addressed your concerns and will incorporate the points mentioned below into the final version. We would be happy to address any further questions you might have.
> ***
> > **Weakness** 1 & **Question** 1: Empirical evidence in support of motivation.
>
> We appreciate the reviewer's concern and agree that empirical evidence is crucial. Our evidence is threefold:
>
> (1) Qualitative analysis of attention distribution. Baseline models exhibit highly concentrated attention, with a few points consistently dominating (lines 609-615). BlindFormer decentralizes attention by revisiting blindspot regions. Rather than discarding the original attention, our blindspot mining and joint optimization refine it to obtain ``a broader and more balanced distribution.``
>
> (2) Quantitative analysis of performance under perturbations. ``Standard Transformers show clear robustness issues in a variety of noisy environments.`` For example, on OBJ-BG with Gaussian noise $\sigma$=0.03, Point-MAE achieves only 47.2% accuracy, while BlindFormer improves it by 13.4%.   This shows that the standard transformer's inherent attention bias exacerbates performance degradation when some regions are noisy or partially missing. Thanks to blindspot mining, BlindFormer encourages the model to exploit a wider range of information cues to solve this problem.
>
> (3) Theoretical justification. Attention adapts gradually and excessive focus leads to sparse features, reducing redundancy and making models brittle. BlindFormer acts as a regularizer, encouraging ``richer and more balanced perceptual representations``, thereby improving generalization and robustness.
> ***
> > **Weakness** 2: Performance improvement.
>
> (1) Consistent improvements across settings. Although individual improvements may seem small, BlindFormer shows consistent improvements over supervised learning and self-supervised backbone models across ``all benchmarks and tasks``, demonstrating its generalizability rather than being tuned for a specific dataset.
>
> (2) Robustness under perturbations. The improvements are more significant under challenging conditions. For example, under Gaussian noise ($\sigma$=0.03), BlindFormer improves the classification performance of Point-MAE on OBJ-ONLY by 17.2%; under rotation perturbations, it improves the segmentation performance of PointGPT-S on ShapeNetPart by 3.3%, demonstrating its value ``in real-world noisy environments.``
> ***
> > **Weakness** 3: Minor writing errors.
>
> Thank you for your careful reading and kind feedback. We have re-reviewed the entire paper and corrected some minor writing errors to ensure clarity and accuracy in the final version.
> ***
> > **Question** 2: The role of perturbation probability.
>
> We would like to clarify the role of dynamic blindspot generation. If masking relies solely on attention scores, it repeatedly targets the same high-attention regions, reducing diversity and causing over-regularization. The perturbation probability adds randomness, ensuring that less-salient regions also have a chance to be masked. This promotes exploration of a wider range of cues. The probability temperature in Table 6(c) in the paper is essentially a balance between the effects of attention probability and perturbation probability. To make it more explicit, the ablation results after removing the perturbation probability are shown in Table A. The consistent drop in performance under this configuration confirms the importance of the perturbation variable in improving generalization and robustness.
>
> Table A. Ablation studies of perturbation probability.
> | Method | OBJ-BG | OBJ-ONLY |
> | ---- | ---- | ---- |
> | Baseline | 47.2 | 37.0 |
> | ABM w/o perturbation | 59.0 | 52.6 |
> | ABM w/ perturbation | 60.6 | 54.2 |
> ***
> > **Question** 3: Symbol explanation.
>
> We apologize for the typo in Eq. 6 — $H^m_i$ should be $H^b_i$, representing the blindspot branch feature. The contrastive loss is a symmetric InfoNCE-style objective between the blindspot ($H^b_i$) and standard representations ($H^s_i$). Its purpose is to align representations learned from complementary attention distributions, encouraging the model to explore blindspot regions while maintaining consistency with the standard view. Thank you for pointing this out, and we will double-check the writing of the paper in the final version.
> ***
> > **Question** 4: Contrastive loss weight.
>
> We would like to explain in detail your doubts about contrastive loss weight. $\lambda$ in Eq. 7 is modulated during training following a two-phase schedule. In the early stage, $\lambda$ is set to 0 so that the model focuses on the original loss $L_{origin}$, ensuring stable representation learning. As training progresses, $\lambda$ is set to 0.6 to incorporate the blindspot-aware contrastive loss $L_{contra}$, enabling the model to align blindspot and standard features. The specific values of $\lambda$ are detailed in the ablation studies Table 6(d), where we show that optimal weighting leads to the trade-off between task performance and blindspot feature alignment.

---

> > ### Comment · Reviewer_oJo8 · 2025-08-02
> > **Follow up discussion**
> >
> > Thank you for answering to my questions.
> >
> > 1. In my understanding, the proposed strategy acts as a "self-regularizer" that helps attention to focus in a mode distributed way across all the points. I there an equivalence between BlindFormer and any other standard regularizing pre-training method?
> >
> > 2. Related to __Question 4__, did you observe a correlation between blindspot feature alignment and solving out-of-distribution tasks? For instance, does focusing on the non-salient regions more would help in scenarios where domain adaptation is needed?

---

> > > ### Author Response · Authors · 2025-08-04
> > > **Further Discussion**
> > >
> > > We sincerely thank the reviewer for reading and responding to our rebuttal. We address their remaining questions below.
> > >
> > > > **Comment** 1: BlindFormer and standard regularizer.
> > >
> > > Thank you for your insightful understanding. In fact, BlindFormer can be viewed as a task-aware form of structured regularization, designed to promote more balanced and robust representations. While this is similar with the overarching goal of standard regularization (i.e., mitigating overfitting to dominant features), it differs notably in both motivation and mechanism.
> > >
> > > (1) Distinct motivation. BlindFormer is specifically designed to address the scarcity of redundant information in point clouds by uncovering and leveraging attentional blindspots. In contrast, standard regularizing methods are typically model-agnostic and not tailored to the data structure.
> > >
> > > (2) Different mechanism. Instead of simply applying generic techniques such as batchnorm, dropout, or weight decay, BlindFormer explicitly encourages the model to attend to low-attention regions, thereby improving attention distribution and enhancing generalization.
> > >
> > > In summary, BlindFormer functions as ``a structured self-regularizer`` that is purposefully aligned with the challenges of sparse 3D data. Its targeted design provides a unique advantage over traditional regularization methods, especially in the field of point cloud Transformers.
> > >
> > > > **Comment** 2: Domain adaptation.
> > >
> > > Thank you for this question. Although our work does not explicitly target domain adaptation, we find that BlindFormer improves generalization under input perturbations, which is conceptually aligned with the goals of domain generalization. For example, under Gaussian noise, categories with weaker original attention show significant accuracy gains (e.g., “display” from 38.10% to 76.19%, “sofa” from 54.76% to 80.95%).  Additionally, BlindFormer better distinguishes between morphologically similar categories (e.g., average accuracy of “desk” and “table” improves from 50.19% to 74.26%), demonstrating its robustness against subtle inter-class ambiguities.
> > >
> > > These observations suggest that BlindFormer enables the model to handle challenging or less typical samples more effectively. This behavior naturally ``enhances resistance to distribution shifts``, as it reduces dependence on domain-specific or spurious correlations. We believe such robustness makes BlindFormer a promising candidate for future exploration in out-of-distribution and domain-adaptive settings.

---

> > > > ### Comment · Reviewer_oJo8 · 2025-08-05
> > > > **Follow up and closing**
> > > >
> > > > Thanks to the reviewers for their promptly answer.
> > > >
> > > > I appreciate the value of the paper, and the new direction in the pre-training for 3D understanding. In an non-static "real world", having domain adaptation capabilities is very important, specially for this particular data modality. I would encourage the authors to add the pertinent additional information suggested by me and the other reviewers in the next version of the paper. I will increase my score to borderline accept.

---

> > > > > ### Author Response · Authors · 2025-08-06
> > > > > **Thanks for suggestions**
> > > > >
> > > > > We sincerely thank the reviewers for recognizing our work's value and constructive suggestions. As you recommended, we will expand the discussion on domain adaptation in the final version to improve the paper's comprehensiveness.
> > > > >
> > > > > Thank you again for your thoughtful review and positive feedback. We are happy to address any further concerns/questions you might have.

---

### Official Review · Reviewer_fgb3 · 2025-06-22

**Clarity:** 3
**Significance:** 3
**Originality:** 3
**Rating:** 4
**Confidence:** 4

**Summary:**

The paper proposes an attentional blindspot mining strategy, which enhances attention to overlooked regions in the point transformer. It also designs a contrastive loss combined with the original loss. The weight of the contrastive loss increases gradually during training. Experiments on various datasets and backbone models demonstrate that the attentional blindspot mining could improve the results.

**Questions:**

$L_{origin}$ is from the original method, e.g., Point-MAE. In ablation studies, how to compute the random blindspot mining and attentional blindspot mining for $L_{origin}$?

Can the strategy of suppressing salient regions be applied to other transformer models, such as image transformers or natural languages?

Suppressing salient regions contradicts the conventional transformer model. How to understand the phenomenon? Are there other strategies to improve transformer?

**Ethical Concerns:**

["NO or VERY MINOR ethics concerns only"]

**Final Justification:**

I appreciate the authors' rebuttal. Most of my concerns are addressed. I choose to keep my score.

**Limitations:**

The limitation is not properly addressed, which should be added. Are there any cases whose accuracy decreases?

**Paper Formatting Concerns:**

N.A.

**Quality:**

3

**Strengths And Weaknesses:**

Strengths:

The attentional blindspot mining enhances the focus on regions overlooked in standard transformer. It seems that the blindspot mining enhances could rectify transformer bias.

The contrastive learning combines previous transformer loss and the proposed blindspot feature, which improves the conventional point transformer model.

Weaknesses:

Though improvements are significant for individual examples, they are small for most examples. I consider the 17.2% increase is an exception, because the original 37%  accuracy is rather low.

---

> ### Author Rebuttal · Authors · 2025-07-31
>
> We thank the reviewer for the positive evaluation and the valuable feedback. We believe that we have fully addressed your concerns and questions, and will incorporate all points mentioned below in the final version. We would be happy to address any further questions you might have.
> ***
> > **Weakness** 1: Performance improvement.
>
> We acknowledge that the +17.2% gain on OBJ-ONLY stems from a challenging baseline scenario, but this does not diminish its significance. The low baseline highlights the brittleness of standard Transformers under noise, where BlindFormer offers ``a substantial robustness improvement.`` Especially, we achieve these gains without architectural modifications or additional inference parameters. BlindFormer demonstrates consistent gains across all datasets and tasks, even when baseline performance is already strong, showing that the approach is not limited to extreme cases. This consistency across diverse conditions underpins the method’s generalizability and practical relevance.
> ***
> > **Question** 1: The original loss in ablation studies.
>
> In our ablation studies, $L_{origin}$ corresponds to loss in the original pre-training method (e.g., the reconstruction loss for Point-MAE). To evaluate the effect of blindspot mining strategies, we incorporate both Random Blindspot Mining (RBM) and Attentional Blindspot Mining (ABM) into BlindFormer’s dual-branch framework. RBM randomly masks a fixed proportion of points to simulate blindspot discovery without attention guidance. In contrast, ABM leverages the attention distribution to suppress salient regions to encourage learning from blindspot cues. They both ensure that the model needs to rely on the features of the mined blindspot region to reconstruct the complete point cloud, and calculate the original loss through $l_2$ Chamfer distance.
> ***
> > **Question** 2: Extension of our approach.
>
> We believe that the strategy of blindspot mining can be generalized to other Transformer-based models, but its impact depends on domain characteristics. Due to the lack of redundant information, over-focusing on a few salient regions is particularly harmful in point clouds. This motivates the emergence of BlindFormer, which reallocates attention to explore blindspot and capture more diverse geometric clues.
>
> In domains such as images or natural language, when redundant information (irrelevant background) significantly deviates from the subject, ``high attention to core semantics is often beneficial.`` When they are somewhat related, occasionally suppressing these regions during training may encourage the model to exploit complementary context. Overall, our method still plays the role of attention regularization. By dynamically mining blindspot regions, the model can better exploit complementary information, thereby improving robustness to occlusion and noisy inputs.
> ***
> > **Question** 3: Phenomenon understanding and strategy improvement.
>
> (1) BlindFormer complements rather than opposes Standard Transformers. Standard self-attention often over-concentrates on a few dominant cues, which can lead to overfitting and reduced robustness, especially for point clouds with limited redundancy. By dynamically masking salient regions, BlindFormer redistributes attention, encouraging integration of complementary under-attended cues for a more balanced representation. Especially, it does not completely negate the original attention bias - ``a more robust attention distribution`` is obtained while preserving the original task and representation contrast.
>
> (2) Other strategies to improve the Transformer. Prior work explores techniques such as attention dropout[1], label pruning[2], and loss-based regularization[3]. BlindFormer offers a principled, data-driven alternative by explicitly leveraging the model’s own attention distribution to identify and learn from blindspots, yielding more generalizable and robust point cloud representations.
>
> [1] Wu, Qiangqiang, et al. "Dropmae: Masked autoencoders with spatial-attention dropout for tracking tasks." CVPR, 2023.
>
> [2] Park, Dongmin, et al. "Robust data pruning under label noise via maximizing re-labeling accuracy." NeurIPS, 2023.
>
> [3] Zhai, Shuangfei, et al. "Stabilizing transformer training by preventing attention entropy collapse." ICML, 2023.
> ***
> > **Limitation** 1: Description of limitations.
>
> We appreciate your suggestions on limitations. As mentioned in lines 624-631, exploration in more tasks and architectures will further demonstrate the generalizability of our method.
>
> Furthermore, we carefully checked all experiments and found that BlindFormer may not improve every category equally. For example, in OBJ-BG classification under Gaussian noise, although overall accuracy improved, a few categories with already highly effective original attention experienced slight drops (e.g., “chair” from 91.03% to 87.18%, “table” from 87.04% to 85.19%), likely due to attention redistribution reducing reliance on original salient cues. However, BlindFormer enables stronger generalization and robustness, leading to significant gains in categories where the original attention struggled (e.g., “bin” from 47.50% to 80.00%, “door” from 7.14% to 57.14%) and overall improved performance.

---

> > ### Comment · Reviewer_fgb3 · 2025-08-05
> >
> > I appreciate the authors' rebuttal. I choose to keep my score.

---

> > > ### Author Response · Authors · 2025-08-06
> > > **Thanks for comment**
> > >
> > > We sincerely thank the reviewer for reading and responding to our rebuttal. We hope our detailed responses have addressed your concerns satisfactorily. We are happy to address any further concerns/questions you might have.

---

### Official Review · Reviewer_yhBh · 2025-06-29

**Clarity:** 3
**Significance:** 3
**Originality:** 3
**Rating:** 5
**Confidence:** 3

**Summary:**

In this paper, the authors propose BlindFormer, which addresses a critical limitation in point cloud Transformers: their tendency to overfocus on salient regions while neglecting less prominent areas. The framework introduces two core innovations:

1. Attentional Blindspot Mining (ABM): Dynamically suppresses high-attention regions during training, forcing the model to explore "blind spots" to capture richer geometric cues.

2. Blindspot-Aware Joint Optimization (BJO): Integrates contrastive learning with the original pretext task, enhancing feature discrimination without sacrificing primary task performance.

Evaluated on benchmarks like ScanObjectNN, ModelNet40, and ShapeNetPart, BlindFormer consistently boosts robustness against noise (Gaussian, rotation, scaling, point dropout) and improves downstream task performance.

**Questions:**

Have you ever tried the method on other tasks?

**Ethical Concerns:**

["NO or VERY MINOR ethics concerns only"]

**Final Justification:**

The authors solved my concerns about the weakness of the paper. I tend to keep my rating as accept.
Moreover, I think the method has potential on other task despite point cloud, such as LLM, MAE or any kinds of task with pretraining-finetuning paradigm.

**Limitations:**

yes.

**Quality:**

3

**Strengths And Weaknesses:**

Strengths:

1. This paper identifies and mitigates attentional bias in point cloud Transformers, a previously underexplored issue critical for 3D understanding.
2. The proposed method, though utilizing existing tools such as topk selection and contrastive learning, is still a novel solution, when previous methods usually add loss on the attention matrix.
3. The method shows significant improvement on three datasets, and is compatible with multiple backbones (Point-MAE, PointGPT-S, Point-BERT) without architectural changes.
4. Code is provided and I believe they will open-source them.

Weakness:

1. The blind spot problem can happen in all transformer backend architectures, especially for the common pre-training-then-finetuning paradigm. The authors should apply their method to other tasks to get more impact.
2. In Equation 6, there is a $H_i^m$. Through the context, I guess that it is the feature of the masked tokens. It is better to explain it clearly in the text. Moreover, since there are two terms in Equation 6, their effect should be ablated in the experiment.
3. When using contrastive loss, we usually want to see the improvement is from positive pairs or both positive and negative pairs. It would be good to provide such ablation study.

---

> ### Author Rebuttal · Authors · 2025-07-31
>
> We thank the reviewer for the positive evaluation and the valuable feedback. We believe that we have fully addressed your concerns and questions, and will incorporate all points mentioned below in the final version. We would be happy to address any further questions you might have.
> ***
> > **Weakness** 1 & **Question** 1: Evaluation on other tasks.
>
> We have evaluated ``object detection`` on ScanNetV2 by replacing 3DETR's encoder with our pre-trained encoder. Although there is a performance gap with methods dedicated to scene understanding, BlindFormer enhances scene comprehension of baseline through dynamic blindspot mining.
>
> Table A. 3D object detection on ScanNet v2.
> | Method | $AP_{25}$ | $AP_{50}$ |
> | ---- | ---- | ---- |
> | Point-MAE | 63.2 | 42.8 |
> | +BlindFormer | 64.1 | 43.5 |
> ***
> > **Weakness** 2: Symbol explanation and effect.
>
> We appreciate the reviewer’s suggestion and agree that the notation could be clearer. $H^m_i$ in Eq. 6 is indeed a typo and should be $H^b_i$, representing the blindspot branch feature.
>
> The two terms in Eq. 6 enforce bidirectional alignment: blindspot to standard and standard to blindspot. Regarding the two terms, we experimented with removing each of them, and the results are reported in the Table B. This bidirectional alignment design ``prevents representation collapse``, ensuring that the standard branch learns from under-attended cues while the blindspot branch remains grounded in the original representation, resulting in richer and more robust features.
>
> Table B. Ablation studies of contrastive loss.
> | Method | OBJ-BG | OBJ-ONLY |
> | ---- | ---- | ---- |
> | Baseline | 47.2 | 37.0 |
> | Blindspot to standard alignment | 59.8 | 53.8 |
> | Standard to blindspot alignment | 58.9 | 52.7 |
> | Bidirectional alignment | 60.6 | 54.2 |
> ***
> > **Weakness** 3: Positive and negative pairs.
>
> We thank the reviewer for the suggestion. Our contrastive loss in Eq. 6 uses both positive and negative pairs, where positive pairs enforce alignment between blindspot and standard branch features, and negative pairs ensure discrimination from other samples. Following your kind suggestion, we conducted an ablation in Table C comparing the use of positive pairs only versus both positive and negative pairs. The results confirms that ``both positive and negative pairs`` are essential for robust blindspot-aware representation learning.
>
> Table C. Ablation studies of positive and negative pairs.
> | Method | OBJ-BG | OBJ-ONLY |
> | ---- | ---- | ---- |
> | Baseline | 47.2 | 37.0 |
> | Positive pairs only | 57.2 | 51.4 |
> | Positive and negative pairs | 60.6 | 54.2 |

---

### Comment · Area_Chair_PSyu · 2025-08-06
**urgent reminder for reviewers to enage with rebuttal**

Dear reviewer

thanks for your contributions to NeurIPS 2025. Following is an urgent reminder if you haven't done so.

Reviewers should stay engaged in discussions, initiate them and respond to authors’ rebuttal, ask questions and listen to answers to help clarify remaining issues. If authors have resolved your (rebuttal) questions, do tell them so. If authors have not resolved your (rebuttal) questions, do tell them so too.

Even reviewer thinks that for some reason there is no need to reply to authors or authors’ rebuttal, please discuss that with author and approve if there is a justified reason or disapprove otherwise.

Please note “Mandatory Acknowledgement” button is to be submitted only when reviewers fulfill all conditions below (conditions in the acknowledgment form): read the author rebuttal engage in discussions (reviewers must talk to authors, and optionally to other reviewers and AC - ask questions, listen to answers, and respond to authors)

Thanks again!

---

### Note · Authors · 2025-08-12

Dear Reviewers and Area Chairs,

We sincerely thank the reviewers for their valuable feedback, which enhances our work. We note that no major flaws were identified, and most questions sought further clarification rather than fundamental changes. As mentioned in our rebuttal and discussion, our method delivers a more balanced and robust attention distribution for point cloud transformers, leading to improved 3D understanding. We summarize the strengths of our paper in four aspects below:

**Originality.** Our work explicitly exploits the attention distribution in point cloud transformers to identify and address attentional blind spots. This mechanism complements standard transformers, yielding consistent gains in both performance and robustness (see our rebuttal to reviewer fgb3).

**Significance.** We propose a blindspot learning framework that mitigates the detrimental effects of attention bias in point cloud transformers. By integrating underutilized yet complementary cues, BlindFormer produces more balanced representations. It is compatible with state-of-the-art backbones without requiring architectural changes(as noted by reviewer yhBh).

**Quality.** The effectiveness of BlindFormer is validated through extensive experiments across multiple benchmarks, demonstrating strong robustness and generalization. Reviewers oJo8 and mAUX acknowledged the soundness of the technical design and the thoroughness of the evaluation.

**Clarity.** We appreciate the reviewers’ recognition of the clear motivation and presentation. We have addressed all questions and will further refine the paper by incorporating their suggestions, including a more explicit discussion of domain adaptation and computational cost analysis.

We once again thank all reviewers for their positive evaluation and valuable suggestions, and the Area Chair for the effort devoted to evaluating our work. We hope these clarifications will be considered in your final assessment.

---

### Decision · Program_Chairs · 2025-09-17

**Decision:**

Accept (poster)

**Comment:**

The authors did a great job in rebuttal and discussion period, addressing most of the concerns by reviewers.

Reviewrs think the paper introduces an interesting perspective for the pre-training of 3D models.

The paper should be revised based upon incorporating the comments, added experiments/analysis, especially method clarity and fairness etc.